# Image Encryption Scheme with Compressed Sensing Based on a New Six-Dimensional Non-Degenerate Discrete Hyperchaotic System and Plaintext-Related Scrambling

**DOI:** 10.3390/e23030291

**Published:** 2021-02-27

**Authors:** Chunyang Sun, Erfu Wang, Bing Zhao

**Affiliations:** Electronic Engineering College, Heilongjiang University, Harbin 150080, China; scy950127@163.com (C.S.); wangerfu@hlju.edu.cn (E.W.)

**Keywords:** non-degenerate chaotic system, compressed sensing, plaintext-related scrambling, two-way diffusion

## Abstract

Digital images can be large in size and contain sensitive information that needs protection. Compression using compressed sensing performs well, but the measurement matrix directly affects the signal compression and reconstruction performance. The good cryptographic characteristics of chaotic systems mean that using one to construct the measurement matrix has obvious advantages. However, existing low-dimensional chaotic systems have low complexity and generate sequences with poor randomness. Hence, a new six-dimensional non-degenerate discrete hyperchaotic system with six positive Lyapunov exponents is proposed in this paper. Using this chaotic system to design the measurement matrix can improve the performance of image compression and reconstruction. Because image encryption using compressed sensing cannot resist known- and chosen-plaintext attacks, the chaotic system proposed in this paper is introduced into the compressed sensing encryption framework. A scrambling algorithm and two-way diffusion algorithm for the plaintext are used to encrypt the measured value matrix. The security of the encryption system is further improved by generating the SHA-256 value of the original image to calculate the initial conditions of the chaotic map. A simulation and performance analysis shows that the proposed image compression-encryption scheme has high compression and reconstruction performance and the ability to resist known- and chosen-plaintext attacks.

## 1. Introduction

As a main carrier of information transmission and storage, the digital image is widely used in people’s lives and in many fields such as education, medical treatment, national defense, and environmental monitoring [1]. If sensitive information is transmitted on an insecure channel, it could be attacked and illegally stolen by criminals. Therefore, the security of image information in transmission and storage has become a focus of attention and research. It is found that traditional ciphers, such as the data encryption standard (DES), international data encryption algorithm (IDEA) and advanced encryption standard (AES) are unfit for image encryption. Therefore, researchers have designed various image encryption algorithms with different techniques. For example, Wu et al. [2] used Sudoku matrices to secretly represent matrix elements and further developed new Sudoku-associated two-dimensional parametric bijection for image scrambling. Zhou et al. [3] proposed an optical multiple image encryption scheme based on discrete cosine transform and nonlinear fractional Mellin transform. The scheme transformed the original image in different spectra based on the use of the discrete cosine transform, and used the nonlinear fractional Mellin transform to the composite spectrum composed of the spectrum of the original (encrypted) image. Other researchers [4,5,6] designed new image encryption and decryption schemes by using elements of the River–Shamir–Adleman (RSA) algorithm combined with bitwise binary operations, quaternary fractional-linear transformations, and projective transformation. Experiments showed that these algorithms are effective for images with clearly defined contours. In addition, there are image encryption schemes using wave functions [7], S-boxes [8,9,10], deoxyrinonucleic acid (DNA) computing [11,12,13], and others, and they all achieve good image security. A chaotic system is a type of nonlinear system that has cryptographic characteristics such as pseudo-randomness and initial value sensitivity. In recent years, many image encryption schemes based on chaotic systems have been proposed [14,15,16,17,18,19]. Based on these, a meaningful plain image is converted into a noise-like or texture-like cipher image, and the size of the cipher image is equal to that of the plain image. However, these encryption algorithms focus on information security and do not take into account the high image data redundancy, strong correlation between pixels, and other characteristics of such images. With the advent of 5G, the amount of data transmitted on networks will grow exponentially. To improve transmission speed and reduce the amount of storage space, it will be necessary to compress images before transmission or storage. Therefore, the aim of this study is to determine how to ensure the security of image encryption while reducing the amount of both data transmission and storage.

Compressed sensing (CS) theory [20,21] is a recent sampling theory that breaks the limitation of the traditional Nyquist sampling theorem and can sample and compress signals simultaneously. If the measurement matrix in the CS framework is regarded as a key, the CS framework can be regarded as a symmetric cryptographic encryption scheme [22]. In recent years, researchers have used CS in the field of image encryption, using the computational security of the measurement matrix to synchronously encrypt and compress the image. For example, [23] proposed an image encryption method based on the combination of orthogonal coding and CS based on double-random phase coding, which compresses all the images into random signals and diffuses them into smooth white noise. In turn, Li et al. [24] proposed a multi-image encryption method based on lifting wavelet transform (LWT), XOR operations, and compressed ghost imaging. In this scheme, the use of line scan compressed ghost imaging improved the reconstruction speed and made high-resolution image compression possible. These image compression and encryption schemes use the whole measurement matrix as the key, which requires a large amount of storage space and is a serious waste of resources.

To solve this problem, some researchers use chaotic mapping to design measurement matrices [25,26,27,28,29]. The initial values and parameters of the chaotic maps are used as keys, which greatly reduces the required storage space. For example, Zhou et al. [25] proposed a method based on CS and the fractional Mellin transform. The author uses a one-dimensional (1D) logistic chaotic map to generate two CS measurement matrices, then measures them in two different directions, and finally uses a nonlinear fractional Mellin transform to obtain the final ciphertext image. Ponuma et al. [26] proposed a new 1D chaotic system, used the resulting chaotic sequence to generate the measurement matrix, and then used the chaotic sequence to scramble and spread the measured values. [27] Chai et al. proposed a visually secure image encryption method based on CS and chaotic maps. In this method, the CS measurement matrix is constructed by oblique tent mapping, where a 1D chaotic map is used to construct the measurement matrix. Because of the low complexity of a low-dimensional chaotic map and its easily predicted trajectory, the randomness of a chaotic sequence generated by this kind of chaotic map is poor. This reduces the performance of image reconstruction and the security of the algorithm [30].

In contrast, high-dimensional chaotic maps, especially hyperchaotic maps, have more variables, parameters, and positive Lyapunov exponents, which is more suitable for image encryption. Therefore, some researchers have proposed encryption schemes that incorporate high-dimensional hyperchaotic systems. Wang et al. [31] proposed a scheme based on tensor CS (TCS) to simultaneously compress and encrypt 3D image sequences. This algorithm uses a non-autonomous Lorenz system to control the generation of three measurement matrices of TCS. Reference [32] proposed an image compression and encryption scheme based on a modified Chen’s hyperchaotic system and 2D CS, in which Chen’s hyperchaotic system constructs a partial Hadamard matrix for use as the measurement matrix, which improves the performance of image compression and reconstruction.

In addition, during CS encryption, because of the linear sampling rule of CS, the information of the plaintext image will be concentrated in each column of the encrypted image, which increases the risk of energy leakage and the possibility that the cryptographic algorithm can be cracked. This reduces the security of image encryption algorithms based on a CS framework [33,34]. To improve security, some researchers add chaotic encryption techniques to image encryption systems based on a CS framework. For instance, Chen et al. [35] proposed a cryptographic algorithm based on CS, which uses a 3D cat mapping to generate the measurement matrix. In addition, chaotic sequences are used in the process of image scrambling and diffusion. The algorithm achieves good compression, but the key in the secondary encryption algorithm is unrelated to the plaintext. Xie et al. [36] proposed a 3D discrete hyperchaotic system to design the measurement matrix and then quantify and Arnold scramble the measurement. Experimental results show that the algorithm has good reconstruction ability, but the algorithm has low correlation with the plaintext images and are vulnerable to known- and chosen-plaintext attacks.

To achieve higher compression ratio and increase the ability of the CS framework to resist known- and chosen-plaintext attacks, this paper proposes a CS encryption scheme based on the new 6D non-degenerate discrete hyperchaotic system. Our main contributions are as follows: (1) a 6D discrete hyperchaotic system with six positive Lyapunov exponents is designed, and the randomness of the generated chaotic sequence is analyzed and applied in the whole scheme. (2) the scheme uses the initial conditions and the SHA-256 value of the plaintext image to calculate the key, which increases the correlation between the algorithm and the plaintext image and improves the scheme’s ability to resist known- and chosen-plaintext attacks. (3) during CS measurement, the Arnold transform is introduced to scramble the image’s sparse coefficient matrix. Next, the hyperchaotic map is used to design the measurement matrix to improve the compression ratio and reconstruction performance of the proposed scheme. (4) a plaintext-related scrambling algorithm and two-way diffusion algorithm are developed to re-encrypt the measurement matrix generated by the compressed sensor. The scrambling process of this algorithm is closely related to the value of the current scrambling pixel, which increases the security of the CS framework. Experimental results and a security analysis show that the scheme has good compression and reconstruction performance. Moreover, it resists known- and chosen-plaintext attacks well.

The rest of this study is organized as follows. In Section 2, a new hyperchaotic system is designed and analyzed. In Section 3, some related knowledge is presented. In Section 4, the image compression and encryption scheme based on 6D discrete hyperchaotic system and CS is introduced in detail. Section 5 shows the experimental results of the proposed scheme and outlines the security analysis. Finally, the last section concludes our work.

## 2. Proposed System and Its Analysis

### 2.1. Six-Dimensional Non-Degenerate Discrete Hyperchaotic System

A non-degenerate high-dimensional hyperchaotic system is a high-dimensional hyperchaotic system with the highest number of positive Lyapunov exponents possible. Under the condition that the chaotic system is globally bounded, increases in both the number and values of the positive Lyapunov exponents increase the complexity of the dynamic behavior of the whole system. Because the structure of a low-dimensional chaotic system is relatively simple, the positive Lyapunov exponent is not large enough, and degeneracy usually occurs. In contrast, the number of positive Lyapunov exponents of a non-degenerate high-dimensional discrete hyperchaotic system is equal to the number of dimensions of the system, and its dynamic behavior is more complex. The trajectory is more unpredictable, and the generated chaotic sequence is more random.

The Jacobi matrix method is commonly used to calculate the Lyapunov exponent of the dynamical system. The Lyapunov exponent is an important basis used to determine the existence of chaotic behavior in dynamical systems. In this study, we use the given positive Lyapunov exponent to construct chaotic system with inverse solutions [37]. The Jacobi matrix method means that the corresponding Lyapunov exponent of the high-dimensional discrete dynamic system is obtained by using the high-dimensional discrete dynamic system, whereas the inverse Jacobi matrix method is used to reconstruct the corresponding high-dimensional discrete dynamic system with the known or Lyapunov exponent given a priori. These two methods focus on the relationship between the Jacobi matrix and its eigenvalues, and the similar matrix is the method used to reconstruct the Jacobi matrix with the eigenvalues of the Jacobi matrix. In this study, a new 6D nondegenerate discrete hyperchaotic system is designed with this method. The specific steps are listed below.

Consider a high-dimensional discrete chaotic system model,
(1)$$S_{n+1}=AS_{n},\;\;n\in N,\;\;S_{n}\in R^{m}$$

1. The six eigenvalues of a given 6 × 6 Jacobi matrix are 3275, 362, 1356, 1013, 627, and 92, respectively. The corresponding rounded Lyapunov exponents are 8.09, 5.89, 7.21, 6.92, 6.44, and 4.52. The diagonal matrix is constructed as follows,
(2)$$\Lambda =\left[\begin{array}{cccccc} 8.09 & 0 & 0 & 0 & 0 & 0\\ 0 & 5.89 & 0 & 0 & 0 & 0\\ 0 & 0 & 7.21 & 0 & 0 & 0\\ 0 & 0 & 0 & 6.92 & 0 & 0\\ 0 & 0 & 0 & 0 & 6.44 & 0\\ 0 & 0 & 0 & 0 & 0 & 4.52 \end{array}\right]$$

2. The nonsingular matrix *q* is set as,
(3)$$q=\left[\begin{array}{cccccc} 2 & 1 & 1 & 1 & 1 & 1\\ 1 & 2 & 1 & 1 & 1 & 1\\ 1 & 1 & 2 & 1 & 1 & 1\\ 1 & 1 & 1 & 2 & 1 & 1\\ 1 & 1 & 1 & 1 & 2 & 1\\ 1 & 1 & 1 & 1 & 1 & 2 \end{array}\right]$$

The inverse matrix *q*^−1^ is then obtained.

3. The rounded parameter matrix *A* is,
(4)$$A=q\Lambda q^{-1}=\left[\begin{array}{cccccc} 9.44 & -0.85 & 0.47 & 0.18 & -0.30 & -2.22\\ 1.67 & 5.36 & 0.79 & 0.50 & 0.20 & -1.90\\ 1.48 & -0.72 & 7.81 & 0.31 & -0.17 & -2.09\\ 1.52 & -0.68 & 0.64 & 7.27 & -0.13 & -2.05\\ 1.59 & -0.61 & 0.71 & 0.42 & 6.38 & -1.98\\ 1.86 & -0.34 & 0.98 & 0.69 & 0.21 & 2.81 \end{array}\right]$$

Therefore, the dynamic Equations of the reconstructed 6D discrete hyperchaotic system are as follows,
(5)$$\left\{ \begin{array}{l} x_{1}\left(i+1\right)=\left(9.44x_{1}\left(i\right)-0.85x_{2}\left(i\right)+0.47x_{3}\left(i\right)+0.18x_{4}\left(i\right)-0.30x_{5}\left(i\right)-2.22x_{6}\left(i\right)\right)\mathrm{mod}1\\ x_{2}\left(i+1\right)=\left(1.67x_{1}\left(i\right)+5.36x_{2}\left(i\right)+0.79x_{3}\left(i\right)+0.50x_{4}\left(i\right)+0.02x_{5}\left(i\right)-1.90x_{6}\left(i\right)\right)\mathrm{mod}1\\ x_{3}\left(i+1\right)=\left(1.48x_{1}\left(i\right)-0.72x_{2}\left(i\right)+7.81x_{3}\left(i\right)+0.31x_{4}\left(i\right)-0.17x_{5}\left(i\right)-2.09x_{6}\left(i\right)\right)\mathrm{mod}1\\ x_{4}\left(i+1\right)=\left(1.52x_{1}\left(i\right)-0.68x_{2}\left(i\right)+0.64x_{3}\left(i\right)+7.27x_{4}\left(i\right)-0.13x_{5}\left(i\right)-2.05x_{6}\left(i\right)\right)\mathrm{mod}1\\ x_{5}\left(i+1\right)=\left(1.59x_{1}\left(i\right)-0.61x_{2}\left(i\right)+0.71x_{3}\left(i\right)+0.42x_{4}\left(i\right)+6.38x_{5}\left(i\right)-1.98x_{6}\left(i\right)\right)\mathrm{mod}1\\ x_{6}\left(i+1\right)=\left(1.86x_{1}\left(i\right)-0.34x_{2}\left(i\right)+0.98x_{3}\left(i\right)+0.69x_{4}\left(i\right)+0.21x_{5}\left(i\right)+2.81x_{6}\left(i\right)\right)\mathrm{mod}1 \end{array}\right.$$

The remainder functions in (5) ensure the boundedness of the system. Because the 6D discrete system (5) has no quadratic and higher-order terms, its Jacobi matrix *J* is a coefficient matrix. Because the coefficient variable matrix A is a rounded approximation, the six Lyapunov exponents in the 6D discrete system (5) need to be checked again. After calculation, the six Lyapunov exponents of system (5) are: $LE_{1}$ = 4.5003, $LE_{2}$ = 8.0696, $LE_{3}$ = 5.9094, $LE_{4}$ = 6.4502, $LE_{5}$ = 7.2205, and $LE_{6}$ = 6.9200, which are very close to the preset values in step 1. When a chaotic system has more than one positive Lyapunov exponent, the chaotic dynamic behavior of the chaotic system is more complex, and it can be called a hyperchaotic system [38]. This 6D discrete chaotic system (5) has six positive Lyapunov exponents, which is the maximum number, so it is a non-degenerate hyperchaotic system. The chaotic attractor phase diagrams of 6D chaotic map are shown in Figure 1 for iterative variables $x_{1}\left(0\right)$ = 0.31, $x_{2}\left(0\right)$ = 0.42, $x_{3}\left(0\right)$ = 0.23, $x_{4}\left(0\right)$ = 0.54, $x_{5}\left(0\right)$ = 0.15, and $x_{6}\left(0\right)$ = 0.66, and 1000 iterations. To calculate these diagrams, 64-bit MATLAB software and double floating-point representation were used.

The chaotic characteristics of the system can be represented by the trajectories of the chaotic attractors in phase space. As Figure 1 shows, the 6D hyperchaotic system has more complex phase space and better dynamic characteristics; the chaotic sequence is also more random. Because the Jacobi matrix of the 6D non-degenerate hyperchaotic system (5) is independent of the output chaotic sequence, the map can obtain a chaotic state for most initial values. In addition, the 6D hyperchaotic system can output six different pseudorandom sequences at the same time, so using the hyperchaotic system in CS compression and encryption can improve the reconstructed image and the security of the compression/encryption scheme.

### 2.2. Approximate Entropy Analysis

Approximate entropy (*ApEn*) measures the probability of generating new patterns in the time series when the dimension changes. The greater the probability of generating new patterns, the more complex the series, and the greater the corresponding approximate entropy [39,40]. In the simulation experiment in this section, we use the method of [41] to calculate the approximate entropy. The specific algorithm for approximate entropy is as follows:

1. Suppose the initial data is the sequence x(1), x(2),…, x(N), and then divide them into m D vectors.
(6)X(i)=[x(i),x(i+1),…,x(i+m−1)]
in which i=1,2,3,…,N−m+1.

2. The distance between x(i) and x(j) is defined as:(7)$$d(i,j)=\underset{k=1-m-1}{\max}\left[\left|x(i+k)-x(j+k)\right|\right]$$

3. Setting a threshold value r(r>0), for each *i*, we can obtain the statistics of d(i,j).
(8)$$C_{i}^{m}(r)=\frac{1}{N-m+1}Sum\left\{d(i,j)<r\right\}$$

4. The mean of logarithm of $C_{i}^{m}(r)$ is written as $\varphi^{m}(r)$ and can be calculated by
(9)$$\varphi^{m}(r)=\frac{1}{N-m+1}\sum_{i=1}^{N-m+1}\ln C_{i}^{m}(r)$$

5. Changing dimension and repeating step 1 to step 4, we can obtain the approximate entropy
(10)$$ApEn(m,r)=\underset{N\to \infty }{\lim}\left[\varphi^{m}(r)-\varphi^{m+1}(r)\right]$$

However, in practical terms, the length of the data sequence is bounded. Therefore, the approximate entropy algorithm is changed into
(11)$$ApEn(m,r,N)=\varphi^{m}(r)-\varphi^{m+1}(r)$$

Pincus [42] found that there exists a minimal dependency between ApEn and *N* when *m* = 2 and r∈[0.1SD(x),0.2SD(x)], where SD(x) is the standard deviation of x. In general, a more complex the time series has a larger approximate entropy. The complexity of the output time series of several hyperchaotic systems and the new 6D chaotic map were evaluated using approximate entropy, and the results are shown in Table 1. These results show that under the same parameters, the output time series of the scheme proposed in this paper has higher complexity.

### 2.3. NIST Test

To check whether a sequence meets the security standard of information encryption, it is necessary to test the randomness of the sequence. At present, most such tests are based on the testing software and standards provided by the National Institute of Standards and

Technology (NIST) [44]. In this study, SP800-22 was selected to test the randomness of the output sequence of the 6D discrete hyperchaotic system. The pseudo-randomness of the sequence is determined by the P value. If the P value is less than 0.01, it means that the tested sequence looks completely non-random. In the experiment, 1000 different random sequences with a length of 10^6^ were tested. Table 2 shows the NIST test results.

The test results show that for the six chaotic sequences generated by the 6D discrete chaotic system, the average passing rate of the P value exceeds 99%, and the passing rate is within an acceptable range, which indicates that the test sequence has good randomness. Therefore, the 6D discrete chaotic system is suitable for encryption algorithms [45].

## 3. CS and Arnold Scrambling

### 3.1. CS

CS [46,47], or compressed sampling, is a novel signal acquisition theory originally proposed by Donoho, Candes, and Tao. According to CS theory, as long as the signal is compressible or sparse in a certain transform domain, the transformed high-dimensional signal can be projected onto a low-dimensional space with an observation matrix independent of the transform basis. As a result, the original signal can be reconstructed with high probability from this small number of projections by solving an optimization problem.

Suppose a measurement matrix Φ that is M×N(M<<N) in size is used to measure a 1D signal *X* of length *N* by CS; that is,
(12)y=Φx

In this way, an 1D observation value *y* of length *M* can be obtained, and the linear projection observation value *y* contains the information needed to reconstruct signal *X*. For the case where the signal *X* itself is not sparse, it can be expressed under a set of orthogonal sparse N×N bases Ψ as follows:(13)x=Ψs

When the number of nonzero coefficients K<<N in vector s, vector s is called K-sparse. The sampling process of CS is expressed as follows:(14)y=ΦΨs=Θs
where Θ=ΦΨ is the sensing matrix.

When vector s is sufficiently sparse and the size of sensing matrix Θ and the measured value satisfies certain conditions, sparse coefficient vector *s* of the original signal can be recovered accurately by solving an optimization problem. That is, by solving the minimized p-norm problem expressed as follows [48]:(15)$$\hat{s}=\arg\min{\left\|s\right\|}_{1}\;s.t.\;y=\Theta s$$
where ${\left\|\cdot \right\|}_{1}$ denotes the $l_{1}$-norm. The $l_{1}$-norm minimization problem is a convex optimization problem, which is equivalent to solving a linear programming problem to recover K-sparse signal $\hat{s}$ and then restoring the original signal using sparse representation $\hat{x}=\Psi \hat{s}$. Some greedy pursuit algorithms such as orthogonal matching pursuit (OMP), matching pursuit, and convex optimization can be used to solve the above Equations.

### 3.2. Arnold Scrambling

In the process of CS, after the plaintext image is decomposed using a sparse basis, the two-dimensional coefficient matrix is obtained. Zhang et al. [49] proved that scrambling the coefficient matrix of the plaintext image under the sparse base and then measuring the CS can effectively relax the limited isometric performance, of the measurement matrix and improve the quality of the reconstructed image. In this paper, the Arnold transform is used to scramble the sparse coefficient matrix.

The Arnold transform is periodic, that is, after many iterations of the transform, the input image will return to its original state. Using the Arnold transform to scramble the sparse coefficient matrix, as the number of iterations increases, the position of non-zero elements in the coefficient matrix will be fully disturbed to achieve the aim of uniform sparsity.

## 4. Image Encryption and Decryption Schemes

The main aims of image compression/encryption scheme proposed in this paper are to improve the performance of image compression reconstruction and encryption security. Therefore, the measurement matrix is designed using a 6D non-degenerate discrete hyperchaotic system to improve the chaotic performance and the Arnold transform is added to the CS process. To improve the ability to resist known- and chosen-plaintext attacks, the measured value matrix is quantified and encrypted, and the scrambling algorithm and bi-directional diffusion algorithm for the plaintext are developed using the hyperchaotic system.

The proposed CS compression/encryption scheme consists of two main processes, CS compression sampling and measurement quantization encryption, as shown in Figure 2. The specific steps are discussed in detail in the following sections.

### 4.1. CS Compression Sampling Process

#### 4.1.1. Key Generation

The secure hashing algorithm SHA-256 is used to obtain the 256-bit hash value of the image. The SHA-256 hash value *H* of a plaintext image is divided into 32 8-bit groups and converted into decimal numbers, so *H* can be expressed as $H=h_{1},h_{2},h_{3},\ldots ,h_{32}$, where $h_{i}$ is a decimal number in the range [0, 255]. The initial values and parameters of the chaotic system are as follows:(16)$$\left\{ \begin{array}{l} x_{1}(0)=\mathrm{mod}((x_{1}{}^{\prime }(0)+(\mathrm{mod}(h_{1}+h_{2}+h_{3}+h_{4}+h_{5},256))/256),1)\\ x_{2}(0)=\mathrm{mod}((x_{2}{}^{\prime }(0)+(\mathrm{mod}(h_{6}+h_{7}+h_{8}+h_{9}+h_{10},256))/256),1)\\ x_{3}(0)=\mathrm{mod}((x_{3}{}^{\prime }(0)+(\mathrm{mod}(h_{11}+h_{12}+h_{13}+h_{14}+h_{15},256))/256),1)\\ x_{4}(0)=\mathrm{mod}((x_{4}{}^{\prime }(0)+(\mathrm{mod}(h_{16}+h_{17}+h_{18}+h_{19}+h_{20},256))/256),1)\\ x_{5}(0)=\mathrm{mod}((x_{5}{}^{\prime }(0)+(\mathrm{mod}(h_{21}+h_{22}+h_{23}+h_{24}+h_{25},256))/256),1)\\ x_{6}(0)=\mathrm{mod}((x_{6}{}^{\prime }(0)+(\mathrm{mod}(h_{26}+h_{27}+h_{28}+h_{29}+h_{30},256))/256),1)\\ r_{1}=r_{1}{}^{\prime }+h_{31}\\ r_{2}=r_{2}{}^{\prime }+h_{32} \end{array}\right.$$
where $\left\{x_{1}{}^{\prime }(0),x_{2}{}^{\prime }(0),x_{3}{}^{\prime }(0),x_{4}{}^{\prime }(0),x_{5}{}^{\prime }(0),x_{6}{}^{\prime }(0)\right\}$ denotes the preset initial values and parameters, $\left\{r_{1}{}^{\prime },r_{2}{}^{\prime }\right\}$ is the pre-iteration number of the chaotic system, and the value interval is [0, 255]. In this way, the chaotic system’s key sets are keys K = $\left\{x_{1}(0),x_{2}(0),x_{3}(0),x_{4}(0),x_{5}(0),x_{6}(0),r_{1},r_{2}\right\}$.

#### 4.1.2. Construction and Optimization of the Chaotic Measurement Matrix

In the measurement process of CS, the element of measurement value *y* is obtained by multiplying the row vector of the measurement matrix by *x*. If the row vectors of the measurement matrix are independent of each other, the correlation among the elements of measured value *y* is reduced accordingly, thus improving the performance of CS reconstruction. Hence, in the scheme proposed in this paper, we orthogonalize the rows of the designed measurement matrix before using it for the CS. The construction and optimization of the measurement matrix consist of the following steps:

1. Given initial values $\left\{x_{1}(0),x_{2}(0),x_{3}(0),x_{4}(0),x_{5}(0),x_{6}(0)\right\}$, execute $r_{1}+r_{2}+MN$ iterations of the hyperchaotic system governed by (1) to obtain six chaotic sequences $\left\{x_{1}(i)\right\}$, $\left\{x_{2}(i)\right\}$, $\left\{x_{3}(i)\right\}$, $\left\{x_{4}(i)\right\}$, $\left\{x_{5}(i)\right\}$, and $\left\{x_{6}(i)\right\}$.

2. Obtain a new chaotic sequence ${\left\{C_{i}\right\}}_{i=0}^{MN-1}$ according to
(17)$${\left\{C_{i}\right\}}_{i=0}^{MN-1}=\mathrm{mod}(x_{1}(i)+x_{2}(i)+x_{3}(i)+x_{4}(i)+x_{5}(i)+x_{6}(i),1)$$

3. Construct measurement matrix Φ column-by-column using chaotic sequence ${\left\{C_{i}\right\}}_{i=0}^{MN-1}$. Matrix $\Phi_{0}$ is expressed as follows: (18)$$\Phi_{0}=\left(\begin{array}{cccc} C_{0} & C_{M} & \cdots & C_{MN-M}\\ C_{1} & C_{M+1} & \cdots & C_{MN-M+1}\\ \vdots & \vdots & \ddots & \vdots \\ C_{M-1} & C_{2M-1} & \cdots & C_{MN-1} \end{array}\right)$$

4. Express matrix $\Phi_{0}$ as $\Phi_{0}={\left(\alpha_{1}{}^{T},\alpha_{2}{}^{T},\ldots ,\alpha_{M}{}^{T}\right)}^{T}$ in the form of row vectors. Let $\beta_{1}=\alpha_{1}$, and use the following formula to normalize the row vectors:(19)$$\beta_{r}=\alpha_{r}-\sum_{i=1}^{r-1}\frac{\langle \beta_{i},\alpha_{r}\rangle }{\langle \beta_{i},\beta_{i}\rangle }\beta_{i}\;r=2,3,\ldots ,M$$
(20)$$\eta_{j}=\frac{\beta_{j}}{\left\|\beta_{j}\right\|}\;j=1,2,\ldots ,M$$
where 〈α,β〉 denotes the inner product of vector *α* and *β*, and ‖β‖ denotes the 2-norm of vector *β*. Finally, measurement matrix $\Phi ={\left(\eta_{1}{}^{T},\eta_{2}{}^{T},\ldots ,\eta_{M}{}^{T}\right)}^{T}$ after orthogonal normalization is obtained.

#### 4.1.3. Image Sparsification and Sparse Coefficient Matrix Scrambling 

A plaintext image *P* with size N×N is sparsified using the discrete wavelet transform. The sparse coefficient matrix, denoted as $SCM_{1}$, is obtained using the following transformation process:(21)$$SCM_{1}=\Psi \times P\times \Psi^{\prime }$$
where Ψ represents the orthogonal wavelet matrix and $\Psi^{\prime }$ represents the transpose of Ψ. The Arnold transform is then used to scramble the elements of coefficient matrix $SCM_{1}$ to obtain $SCM_{2}$.

#### 4.1.4. Compression Sampling

Measurement matrix Φ is used to measure coefficient matrix $SCM_{2}$ column by column, and the measured value matrix $C_{0}$ with a size of M×N is obtained. At this time, the amount of data in $C_{0}$ is significantly less than that of the original image, which is convenient for the subsequent encryption process.

### 4.2. Measurement Quantization Encryption

#### 4.2.1. Measurement Matrix Quantization

The value of the output matrix after CS measurement generally extends beyond the grayscale range of [0, 255]. Therefore, it must be quantized before encryption. This is essentially a linear scaling and rounding of the measured value, limiting it to the range of [0, 255]. The quantization process is expressed as follows:(22)$$P_{0}=floor\left(\frac{255\times (C_{0}-\min)}{\max-\min}\right)$$
where floor(x) represents the maximum integer value that is not greater than *x* and min and max are the minimum and maximum elements, respectively, of measurement matrix $C_{0}$.

#### 4.2.2. Generation of the Cryptographic Matrix

Four cryptographic matrices are generated using the proposed 6D discrete hyperchaotic system. Using the same chaotic sequence used in Section 4.1.2 improves the utilization rate of the hyperchaotic sequences and the overall efficiency of the algorithm. The steps are as follows:The initial values $\left\{x_{1}(0),x_{2}(0),x_{3}(0),x_{4}(0),x_{5}(0),x_{6}(0)\right\}$ in key *K* are used to iterate the 6D discrete hyperchaotic system $r_{1}+r_{2}$ times. Next, the system continues to iterate *MN* times to obtain six chaotic sequences, which are recorded as $\left\{x_{1}(i)\right\}$, $\left\{x_{2}(i)\right\}$, $\left\{x_{3}(i)\right\}$, $\left\{x_{4}(i)\right\}$, $\left\{x_{5}(i)\right\}$, and $\left\{x_{6}(i)\right\}$, where i=1,2,…,MN.The following Equations are used to generate cryptographic matrices *X*, *Y*, *Z*, and *W* in turn:(23)$$\left\{ \begin{array}{l} X(m,n)=floor(\mathrm{mod}((x_{1}((m-1)\times N+n)+x_{2}((m-1)\times N+n)+x_{3}((m-1)\times N+n))\times 10^{15},256))\\ Y(m,n)=floor(\mathrm{mod}((x_{2}((m-1)\times N+n)+x_{3}((m-1)\times N+n)+x_{4}((m-1)\times N+n))\times 10^{15},256))\\ Z(m,n)=floor(\mathrm{mod}((x_{3}((m-1)\times N+n)+x_{4}((m-1)\times N+n)+x_{5}((m-1)\times N+n))\times 10^{15},256))\\ W(m,n)=floor(\mathrm{mod}((x_{4}((m-1)\times N+n)+x_{5}((m-1)\times N+n)+x_{6}((m-1)\times N+n))\times 10^{15},256)) \end{array}\right.$$
where m=1,2,…,M,n=1,2,…N. Matrices *X* and *Y* are used for diffusion and matrices *Z* and *W* are used for scrambling.

#### 4.2.3. Forward Diffusion

Measurement matrix $P_{0}$ is transformed into matrix *A* by forward diffusion. The specific steps are as follows:

1. The first row of measurement matrix $P_{0}$ is diffused from left to right as follows:(24)$$\left\{ \begin{array}{l} A(1,1)=(P_{0}(1,1)+X(1,1)+r_{1})\mathrm{mod}256\\ A(1,j)=(P_{0}(1,j)+A(1,j-1)+X(1,j))\mathrm{mod}256\;j=2,3,\ldots N \end{array}\right.$$

2. The first column of measurement matrix $P_{0}$ is diffused from top to bottom as follows:(25)$$A(i,1)=(P_{0}(i,1)+A(i-1,1)+X(i,1))\mathrm{mod}256\;i=2,3,\ldots M$$

3. Measurement matrix $P_{0}$ is diffused in the first round along the diagonal direction as follows:(26)$$A(i,j)=(P_{0}(i,j)+A(i-1,j)+A(i,j-1)+X(i,j))\mathrm{mod}256\;i=2,3,\ldots M,j=2,3,\ldots ,N$$

After the above diffusion operation, matrix *A* is obtained.

#### 4.2.4. Plaintext-Related Scrambling

Cryptographic matrices *Z* and *W* are used to calculate the permutation position coordinates A(m,n) of matrix A(i,j), and then switch the positions of A(i,j) and A(m,n). The steps for scrambling are as follows:

1. Calculate the sums of the row and column of A(i,j) (excluding element A(i,j)), respectively denoted as $Sum_{row}(i)$, $Sum_{col}(j)$, that is,
(27)$$\left\{ \begin{array}{l} Sum_{row}(i)=sum(A(i,1\;to\;N))-A(i,j)\\ Sum_{col}(j)=sum(A(1\;to\;M,j))-A(i,j) \end{array}\right.$$

2. Permutation coordinate A(m,n) of matrix A(i,j) is calculated as follows:(28)$$\{\begin{array}{l} m=(Z(i,j)+Sum_{row}(i))\mathrm{mod}M\\ n=(W(i,j)+Sum_{col}(j))\mathrm{mod}N \end{array}$$

3. If the calculation is m=i or n=j, the permutation position coordinates of the element are recalculated according to
(29)$$\{\begin{array}{l} m=(i^{2}\mathrm{mod}M)+1\\ n=(j^{2}\mathrm{mod}N)+1 \end{array}$$

Using the above steps, the first row of matrix *A* is scrambled, then the first column of matrix *A* is scrambled, and then the elements of matrix *A* are scrambled in order from left to right and from top to bottom. The scrambled matrix is denoted as *B*.

#### 4.2.5. Backward Diffusion

Matrix *B* is transformed into matrix $C_{1}$, which is the cipher-text image, by backward diffusion. The steps are as follows:

1. Diffuse the last row of matrix *B* from right to left using the following expression:(30)$$\left\{ \begin{array}{l} C_{1}(M,N)=(B(M,N)+Y(M,N)+r_{2})\mathrm{mod}256\\ C_{1}(M,j)=(B(M,j)+C_{1}(M,j+1)+Y(M,j))\mathrm{mod}256\;j=N-1,\ldots ,2,1 \end{array}\right.$$

2. Diffuse the last column of matrix *B* from the bottom up as follows:(31)$$C_{1}(i,N)=(B(i,N)+C_{1}(i+1,N)+Y(i,N))\mathrm{mod}256\;i=M-1,\ldots ,2,1$$

3. Matrix *B* is diffused in the second round along the inverse diagonal direction by
(32)$$C_{1}(i,j)=(B(i,j)+C_{1}(i+1,j)+C_{1}(i,j+1)+Y(i,j))\mathrm{mod}256\;i=M-1,\ldots ,2,1,j=N-1,\ldots ,2,1$$

After the above operation, cipher-text image $C_{1}$ is obtained.

### 4.3. Image Decompression and Decryption Process

The decompression and decryption process is the reverse of the compression and encryption process. The steps are summarized as follows:Six key streams are obtained by iterating the new 6D discrete hyperchaotic system with key K, and then four cryptographic matrices are obtained.The received cipher-text image is processed using the inverse operations of backward diffusion, scrambling and forward diffusion, and quantized measured value matrix $P_{0}$ is obtained.Recovered measurement $C_{0}$ is obtained by inverse quantization of quantized measurement matrix $P_{0}$ according to
(33)$$C_{0}=\frac{P_{0}\times (\max-\min)}{255}+\min$$Key K is used to iterate the new 6D discrete hyperchaotic system and obtain chaos measurement matrix Φ.Restored measurement matrix $C_{0}$ is reconstructed using the OMP algorithm, and restored sparse coefficient matrix $SCM_{2}$ scrambled by the Arnold transform is obtained.The inverse Arnold transformation is performed on coefficient matrix $SCM_{2}$ to obtain restored sparse coefficient matrix $SCM_{1}$.To obtain the restored original image *P* of size N×N, a sparse inverse transformation is performed on $SCM_{1}$ according to
(34)$$P=\Psi^{\prime }\times SCM_{1}\times \Psi$$

## 5. Simulation and Performance Analysis

To illustrate the performance of the proposed scheme, simulation results and a performance analysis are presented in this section. In this study, MATLAB R2016a was used for the simulation experiments. Because the difference in the pixel distributions of different images will affect the compression performance and encryption results of the algorithm, we use different grayscale images of size 256×256 for the simulation tests such as Lena, Cameraman, Goldhill, and Clock. The compression ratio of the images was set to 50%, the discrete wavelet transform was used to sparsify the image, and the OMP algorithm was used to reconstruct the image. The initial values and parameter settings of the 6D discrete hyperchaotic system were as follows: $x_{1}{}^{\prime }(0)$ = 0.276025076998578, $x_{2}{}^{\prime }(0)$ = 0.585267750979743, $x_{3}{}^{\prime }(0)$ = 0.547215529963803, $x_{4}{}^{\prime }(0)$ = 0.489764395788231, $x_{5}{}^{\prime }(0)$ = 0.694828622975817, $x_{6}{}^{\prime }(0)$ = 0.840717255983663, $r_{1}{}^{\prime }$ = 41, and $r_{2}{}^{\prime }$ = 136. The parameters of the Arnold transformation were set to p=3, q=5, and n=10.

The simulation results are shown in Figure 3. The first to fourth columns represent the original images, the encrypted images based on CS, the final encrypted images, and the decrypted reconstructed images. From the perspective of the final encryption results, no meaningful information can be obtained from the ciphertext image in the third column. The compression and reconstruction results show that the size of the ciphertext image is only half that of the original image, which greatly reduces the image file size, storage space, and transmission bandwidth required for transmission. Moreover, as the fourth column shows, the reconstructed images are visually very similar to the corresponding original images.

These results demonstrate the encryption results and compression performance of the scheme well. To comprehensively evaluate the compression performance and encryption security of the proposed scheme, a detailed analysis was carried out to evaluate the compression performance, statistical properties, key space, key sensitivity, plaintext sensitivity, and information entropy. The results are presented in the folowing sections.

### 5.1. Image Decompression and Decryption Process

Figure 4 compares the original and reconstructed images for different compression ratios. The compression ratios from the second to fourth columns are 25%, 50%, and 75%. The figure shows that when the compression ratio is 75%, the difference between the reconstructed image and the original image is very small. As the compression ratio decreases, the quality of the reconstructed image gradually decreases, but even when the compression ratio is 25%, the main information of the original image can still be distinguished in the reconstructed image.

By evaluating the PSNR and SSIM of the reconstructed image under different compression ratios, the reconstruction quality of the image can be better understood. PSNR is the most widely used image objective evaluation index. PSNR is usually used to judge the quality of the decrypted image [50]. A larger PSNR value indicates a smaller the difference between the two images and a higher reconstruction accuracy. PSNR is defined as follows:(35)$$PSNR=10\mathrm{lg}\frac{255\times 255}{\left(\frac{1}{M\times N}\right)\sum_{i=1}^{M}\sum_{j=1}^{N}{\left(X(i,j)-Y(i,j)\right)}^{2}}.$$

SSIM is also a commonly used image quality evaluation index. It measures image similarity with respect to three factors: brightness, contrast, and structure [51]. The value range of SSIM is [0, 1], and a larger value indicates a smaller image distortion. SSIM is defined as follows:
(36)$$SSIM=\frac{\left(2\mu_{x}\mu_{y}+c_{1}\right)\left(2\sigma_{xy}+c_{2}\right)}{\left(\mu_{x}^{2}+\mu_{y}^{2}+c_{1}\right)\left(\sigma_{x}^{2}+\sigma_{y}^{2}+c_{2}\right)},$$
where $\mu_{x}$ and $\mu_{y}$ are the means of the plaintext and ciphertext images, respectively; $\sigma_{x}$ and $\sigma_{y}$ are their variances, and $\sigma_{xy}$ is their covariance. Here, $c_{1}={\left(k_{1}L\right)}^{2}$, $c_{2}={\left(k_{2}L\right)}^{2}$, $k_{1}=0.01$, $k_{2}=0.03$, and L=255.

It can be seen from Figure 5 that as the compression ratio of the ciphertext image increases, the SSIM and PSNR values of the reconstructed image also increase. Moreover, the PSNR values of the reconstructed image are all above 30 dB, even at a compression rate of 30% for the four images tested. PSNR values over 30 dB indicate that the compression/encryption scheme proposed in this article achieves a very high compression rate and good reconstruction results. In addition, when the compression rate is fixed, different images have different PSNR and SSIM values, so the compression ratio can be flexibly set for different types of images according to requirements.

The following analysis compares the compression/encryption scheme proposed in this article with other hyperchaotic system-based schemes. As an example, Table 3 shows the PSNR values of Lena images reconstructed with different schemes when the compression ratio is 0.75.

It can be seen from Table 3 that of the compression/encryption schemes are based on different hyperchaotic systems, the PSNR values of the image recovered by the compression/encryption scheme based on the proposed 6D hyperchaotic system is relatively large, indicating that it has good compression and reconstruction performance.

### 5.2. Statistical Analysis

#### 5.2.1. Correlation Analysis

The correlation between adjacent pixels is a very important performance indicator for evaluating image encryption schemes. Because of the characteristics of image data, there is a strong correlation between adjacent pixels in an image, so the correlation coefficient of plaintext images is usually very high. An effective image encryption scheme should minimize the correlation of ciphertext images, thereby improving its ability to resist statistical attacks [53]. The closer the correlation coefficient of adjacent pixels is to 0, the better the encryption results. The correlation coefficient is calculated as follows:(37)$$\left\{ \begin{array}{l} \mathrm{cov}(x,y)=\frac{1}{N}\sum_{i=1}^{N}\left(x_{i}-E\left(x\right)\right)\left(y_{i}-E\left(y\right)\right)\\ E\left(x\right)=\frac{1}{N}\sum_{i=1}^{N}x_{i}\\ D\left(x\right)=\frac{1}{N}\sum_{i=1}^{N}{\left(x_{i}-E\left(x\right)\right)}^{2}\\ r_{xy}=\frac{\mathrm{cov}\left(x,y\right)}{\sqrt{D\left(x\right)D\left(y\right)}} \end{array}\right.$$

In (37), *x* and *y* are the pixel values of the adjacent pixels of the image to be tested, *N* is the number of selected pixels, and E(x) and E(y) are the average values of all *x* and *y*. In addition, D(x) is the variance, cov(x,y) is the covariance, and $r_{xy}$ is the correlation coefficient. In this experiment, we used Lena, Cameraman, Peppers, and Clock, with a size of 256 × 256, as the plaintext images. To calculate the correlation coefficient, 10,000 pairs of pixels were randomly selected from the plaintext image and ciphertext images in the horizontal, vertical, and diagonal directions. The calculation results are shown in Table 4.

The data in Table 4 show that the correlation coefficients of adjacent pixels in the plaintext image are all close to 1 in all three directions, whereas the corresponding correlation coefficients in the encrypted image are all close to 0, which indicate that the adjacent pixels of the ciphertext image have almost no correlation between adjacent pixels. Hence, the encryption algorithm proposed in this paper can effectively reduce the correlation between the adjacent pixels of encrypted images. As a result, attackers cannot obtain useful information from encrypted images through statistical attacks.

To compare the correlation between adjacent pixels of the image before and after encryption more intuitively, we use the Lena image as an example and plot the correlation of their adjacent pixels in the horizontal, vertical, and diagonal directions in Figure 6. This figure shows the pixel distributions of the Lena image before and after encryption, and it can be seen that the pixel distributions follow a certain rule. After encryption by the proposed encryption algorithm, the distribution of the adjacent pixels of the ciphertext image becomes uniform, and the correlation is substantially reduced. Table 5 shows the correlation results of adjacent pixels in different directions for each evaluated scheme. When compared with existing methods [29,54], the proposed encryption scheme yields images with weaker correlation in the horizontal and vertical directions.

#### 5.2.2. Histogram Analysis

The image histogram is often used to count the frequencies of each pixel value, and it can intuitively display the distribution of all the image pixel values. To prevent an attacker from obtaining useful statistical information from the ciphertext histogram, the encrypted images corresponding to different plaintext images should have distributions similar to the distribution of white noise. Figure 7 shows the simulation results of the histograms.

It can be seen from Figure 7 that the pixel value distributions of different plaintext images have their own characteristics and their ciphertext image histograms are all approximately uniformly distributed. Moreover, the attacker cannot obtain any statistical information from the ciphertext image. Therefore, the image compression/encryption algorithm proposed in this paper can resist statistical analysis attacks.

### 5.3. Information Entropy Analysis

Information entropy effectively measures the randomness of image information. Generally, higher image information entropy indicates higher uncertainty. For encrypted images, a larger ciphertext image information entropy indicates better image encryption. The information entropy is calculated as follows:(38)$$H_{x}=-\sum_{i=0}^{2^{N}-1}P(x_{i})\log_{2}P(x_{i})$$
where $P(x_{i})$ represents the probability of the grayscale value $x_{i}$ and N represents the number of bits of $x_{i}$. For a grayscale random image with a gray level of 256, the theoretical value of $H_{x}$ is 8 [55]. Table 6 shows the information entropy of images encrypted using different compression rates.

Table 6 shows that the information entropy of each ciphertext image is very close to the theoretical value even with different compression rates. Therefore, the compression/encryption scheme proposed in this paper can achieve image compression and encryption simultaneously, and the encrypted image has good randomness and sufficient security.

### 5.4. Key Space Analysis

The key space reflects the ability of the image encryption algorithm to resist brute force attacks. A larger key space indicates that an algorithm will perform better under brute force attacks. In cryptography, to ensure the security of the encryption system, the key space must be larger than $2^{100}$ [56]. In the encryption scheme proposed in this paper, the key is composed of the initial value of chaos and the number of initial iterations, that is, Key = $\left\{x_{1}(0),x_{2}(0),x_{3}(0),x_{4}(0),x_{5}(0),x_{6}(0),r_{1},r_{2}\right\}$, with a total of eight parameters, where $\left\{r_{1},r_{2}\right\}$ is a 9-bit integer with a value range of [0, 511]. According to the IEEE floating-point standard [57], the precision of a double-precision number is about $10^{-15}$, so our key space is up to $Key_{total}={\left(2^{9}\right)}^{2}\times {\left(10^{15}\right)}^{6}\approx 2^{318}$ in size. If the parameters and iterations of the Arnold transform are also regarded as keys, our key space is even larger. Therefore, the key space of the algorithm meets the standard for security and can resist exhaustive attacks.

### 5.5. Key Sensitivity Analysis

Key sensitivity is an important indicator that evaluates the security performance of a cryptographic system. In a good cryptographic system, using two groups of slightly different keys to encrypt the same plaintext should produce two completely different ciphertext images; similarly, for the same ciphertext image, if the key is only slightly different from the correct key for decryption, the decrypted image should also be completely different.

#### 5.5.1. Key Sensitivity Analysis of Encryption Process

In the key sensitivity analysis, a cipher image is obtained by encrypting Lena image with a secret key $x_{1}(0)=0.813939566318005$, $x_{2}(0)=0.734699782001722$, $x_{3}(0)=0.556216746791949$, $x_{4}(0)=0.914120914440195$, $x_{5}(0)=0.762839996440578$, $x_{6}(0)=0.656111356786732$, $r_{1}=200$, and $r_{2}=200$. Taking $x_{1}(0)$ as an example, the key $x_{1}{}^{\prime }(0)$ is obtained by introducing a tiny change ($10^{-15}$), and the Lena image is encrypted with this key. The results are shown in Figure 8.

Figure 8 shows the difference between the corresponding cryptographic image and the two cryptographic images obtained by encrypting the same image with the two sets of keys. The results show that a small change in the key will lead to a completely different encrypted image.

#### 5.5.2. Key Sensitivity Analysis of the Decryption Process

To evaluate the key sensitivity of the decryption process, we used the above key to encrypt the Lena image. Taking $x_{1}(0)$, $x_{2}(0)$ as an example, we decrypted the Lena image by introducing a small change ($10^{-15}$) to obtain key $x_{1}{}^{\prime }(0)$, $x_{2}{}^{\prime }(0)$. The result is shown in Figure 9.

Figure 9 shows the decrypted images obtained by decryption using the correct key and an incorrect key with minor changes. The results show that any slight change in the decryption key will cause the decryption to fail.

These simulation results demonstrate that the image compression/encryption algorithm proposed in this paper has good key sensitivity.

### 5.6. Plaintext Sensitivity Analysis

In known- or chosen-plaintext attacks, the attacker uses the same key to encrypt two images after making small changes to a certain plaintext image. By comparing the two encrypted images, he or she can determine the difference between the plaintext image and the encrypted image to crack the password system. To resist this type of attack, the encryption system should have strong plaintext sensitivity, that is, two plaintext images with small differences should have substantial differences after passing through the encryption system.

In the compression/encryption scheme based on 6D discrete chaotic system proposed in this paper, the SHA-256 function is used to calculate the hash value of the plaintext image, and it is used as the initial value of the hyperchaotic system. When a plaintext image pixel changes slightly, a completely different chaotic sequence will be output, which leads to a completely different measurement matrix and cryptographic matrix for the encryption phase. Similarly, when scrambling the plaintext, the position of each pixel after scrambling is closely related to its pixel value, and if a bit of the pixel value changes, the corresponding scrambled position will also change. The attacker is unable to derive the key used by this algorithm from any encrypted image, so it is unable to decrypt the plaintext image. Therefore, the compression/encryption scheme proposed in this paper can resist both known- and chosen-plaintext attacks.

To evaluate the plaintext sensitivity of the encryption scheme proposed in this paper, we quantitatively analyzed the difference between the ciphertext images of two slightly different plaintext images. The main indicators to measure the difference between the two images include the pixel change rate (NPCR) and the unified average change intensity (UACI) [58]. The former reflects the ratio of the number of different pixels to the number of all pixels, whereas the latter reflects the average ratio of the difference of all the corresponding pixels to 255 in the two images. Typical values of NPCR and UACI are 99.61% and 33.46%, respectively, and these metrics are calculated as follows:(39)$$\{\begin{array}{c} D(i,j)=\{\begin{array}{l} 0,C_{1}(i,j)=C_{2}(i,j)\\ 1,C_{1}(i,j)\neq C_{2}(i,j) \end{array}\\ NPCR(C_{1},C_{2})=\frac{1}{M\times N}\sum_{i=1}^{M}\sum_{j=1}^{N}D(i,j)\times 100\%\\ UACI(C_{1},C_{2})=\frac{1}{M\times N}\sum_{i=1}^{M}\sum_{j=1}^{N}\frac{\left|C_{1}(i,j)-C_{2}(i,j)\right|}{255}\times 100\% \end{array}$$

We choose the 256 × 256 Lena, Cameraman, Peppers, and Clock images as the experimental images. We randomly selected a pixel $C_{1}(i,j)$, changed its pixel value by an increment of 1 to obtain image $C_{2}(i,j)$. Using the same key to encrypt the two images, we calculated the NPCR and UACI. The results are shown in Table 7.

Table 7 shows that the calculated NPCR and UACI values are very close to the ideal values, indicating that the encryption scheme proposed in this paper has strong plaintext sensitivity.

### 5.7. Time Complexity Analysis

Except for security, time is also an index used to assess the performance of image encryption algorithms. The experimental environment of this study included MATLAB R2016a, Inter Core i5-3230M with a central processing unit (CPU) at a frequency of 2.6 GHz with 8-GB random access memory (RAM) on Windows 10. The aim was to analyze the encryption and decryption time of different images at different CRs. The experiment was repeated 20 times, and the results of the above experiments were then averaged. The results are shown in Table 8.

The following can be observed from Table 8. (1) Different plaintext images have minor effects on the encryption and decryption times. (2) For the same original image, the encryption and decryption time at different compression ratios are different, and as the CR increases, the decryption time also increases. The main reason is attributed to the fact that in the CS reconstruction process used to solve the optimal solution, the time needed to complete the process increases as the measurement matrix increases. In addition, the SHA-256 algorithm was used to generate the hash value of a plaintext image and further calculate the initial value of a chaotic system to improve the ability to resist the chosen-plaintext attacks. However, owing to the structure of the algorithm and the large number of iterations, the encryption speed is reduced. To address the efficiency problem of the hash algorithm, relevant research has been conducted on adaptive chaotic maps [59,60,61], including the use of an adaptive chaotic map to generate the hash function [59]. In this scheme, the adaptive map is used as the compression function, whereas in the piecewise adaptive maps, different adaptive coefficients can be flexibly used according to different subdomains. The results from previous studies indicate that the hash function constructed by the adaptive chaotic map can be used as one of the effective methods for solving the time costs of an encryption system. In future research, we will consider different hash function construction methods, and we will introduce them into the compression encryption system proposed in this paper for in-depth research, considering the complexity of the entire system while optimizing the time performance of this scheme.

In summary, the encryption time of the scheme proposed in this study is shorter, and the larger the compression rate, the longer the decryption time. Therefore, the algorithm performs better at the smaller compression rate. Therefore, in practical applications, a comprehensive selection of encryption time and CR should be made according to the actual situation. The decryption speed of this scheme is slow. In the follow-up research, block CS or other reconstruction algorithms can be considered to improve the encryption and decryption speeds. Moreover, the encryption and decryption times depend on the CPU, RAM size, operating system and programming language, and the efficiency of encryption and decryption can be improved by optimizing the selection of the above parameters.

## 6. Conclusions

In this paper, a 6D non-degenerate discrete hyperchaotic system with six positive Lyapunov exponents was proposed and used in a CS encryption framework. In addition, a CS encryption scheme that can realize both image compression and encryption was proposed. We also confirmed that the analysis and verification of the fact that the 6D discrete hyperchaotic system proposed herein was feasible and can be used for image encryption. In the encryption scheme, the Arnold transform is used to scramble the sparse coefficients to obtain uniform sparsity. Then, the measurement matrix is designed and optimized using the proposed 6D hyperchaotic system, and finally, the CS measurement process is carried out. In the measured value encryption stage, the position of the scrambled element is closely related to the value of the element, which has high plaintext and key sensitivity, and can resist known- and chosen-plaintext attacks. In addition, the key used in the chaotic system is generated by the SHA-256 value of the original image, which greatly increases the ability of the algorithm to resist plaintext attacks. Simulation experiments and analysis show that the CS encryption scheme proposed in this paper has better compression and reconstruction performance as well as the ability to resist known- and chosen-plaintext attacks.

However, the study had the following limitations:(1)In practical applications, color images or high-resolution images may be more extensively used. However, this scheme only realizes the compression and encryption of grayscale images.(2)The decryption time of this scheme is relatively long.(3)The scheme only compresses and encrypts one image at a time.

Therefore, to overcome these limitations of the current scheme, future work will focus on the:(1)extension of the scheme to color images or high-resolution images,(2)consideration using block compressed sensing or other reconstruction algorithms to improve the speed of encryption and decryption, and(3)simultaneous realization of the compression and encryption of multiple images.

## Figures and Tables

**Figure 1 entropy-23-00291-f001:**
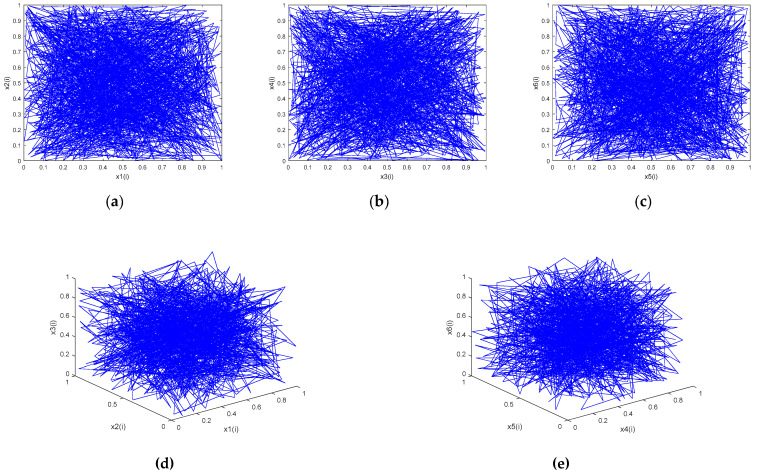
Phase diagrams of chaotic attractors. (**a**) $x_{1}\left(i\right)-x_{2}\left(i\right)$; (**b**) $x_{3}\left(i\right)-x_{4}\left(i\right)$; (**c**) $x_{5}\left(i\right)-x_{6}\left(i\right)$; (**d**) $x_{1}\left(i\right)-x_{2}\left(i\right)-x_{3}\left(i\right)$; and (**e**) $x_{4}\left(i\right)-x_{5}\left(i\right)-x_{6}\left(i\right)$.

**Figure 2 entropy-23-00291-f002:**
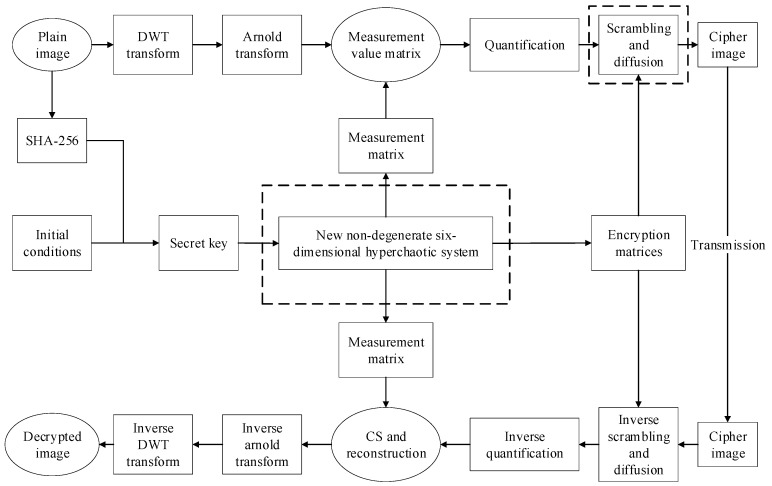
Image encryption and decryption processes in the proposed scheme.

**Figure 3 entropy-23-00291-f003:**
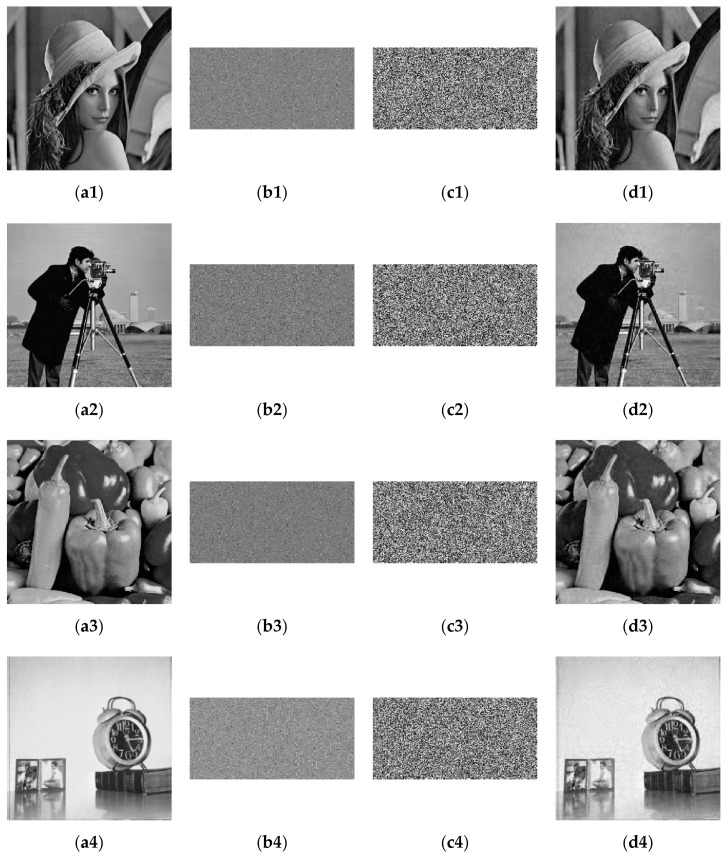
Simulation results. (**a1**–**a4**) Plaintext images; (**b1**–**b4**) CS encrypted images; (**c1**–**c4**) Ciphertext images; (**d1**–**d4**) Decrypted images.

**Figure 4 entropy-23-00291-f004:**
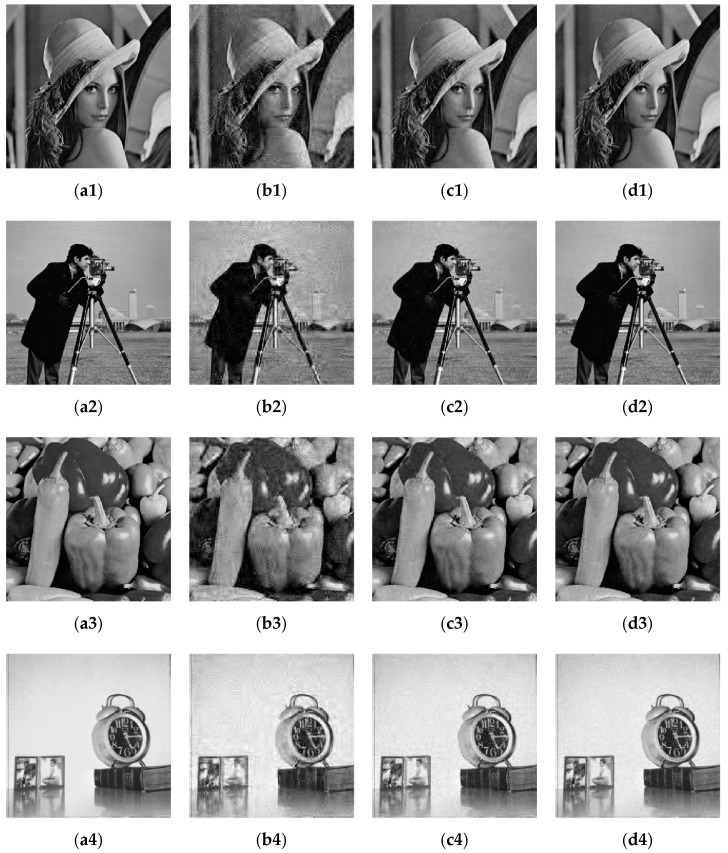
Reconstructed results of images compressed with different compression ratios. (**a1**–**a4**) Plaintext images; (**b1**–**b4**) Reconstructed images when the compression ratio is 0.25; (**c1**–**c4**) Reconstructed images when the compression ratio is 0.5; (**d1**–**d4**) Reconstructed images when the compression ratio is 0.75.

**Figure 5 entropy-23-00291-f005:**
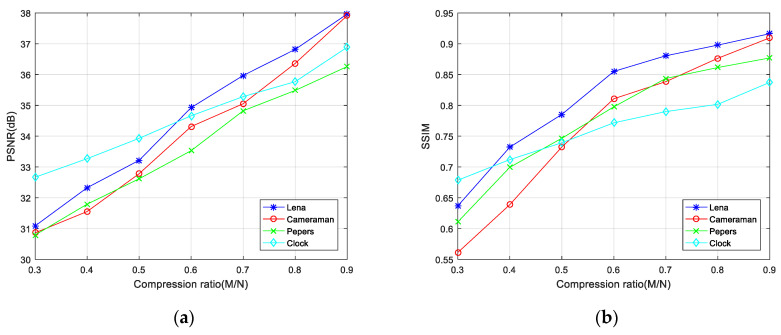
Compression ratio and image reconstruction quality. (**a**) PSNR of reconstructed images under different compression ratios; (**b**) SSIM of reconstructed images under different compression ratios.

**Figure 6 entropy-23-00291-f006:**
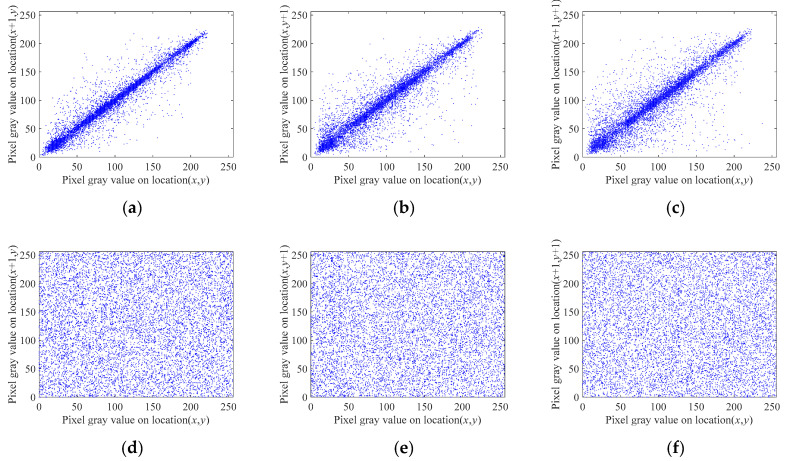
Distribution of adjacent pixels. (**a**) Plaintext horizontal adjacent pixels; (**b**) Plaintext vertical adjacent pixels; (**c**) Plaintext diagonal adjacent pixels; (**d**) Ciphertext horizontal adjacent pixels; (**e**) Ciphertext vertical adjacent pixels; (**f**) Ciphertext diagonal adjacent pixels.

**Figure 7 entropy-23-00291-f007:**
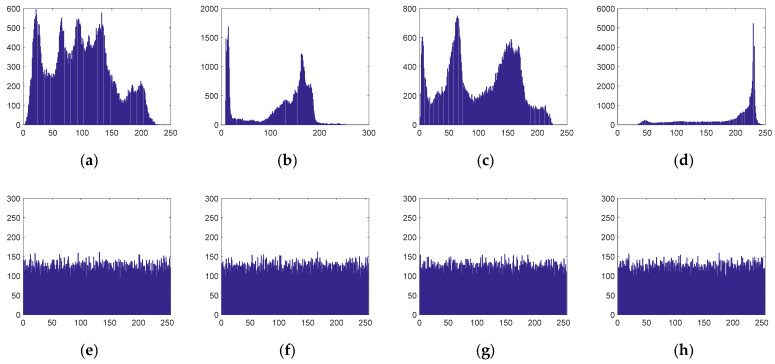
Histograms of the original and encrypted images. (**a**) Plaintext histogram of Lena; (**b**) Plaintext histogram of Cameraman; (**c**) Plaintext histogram of Peppers; (**d**) Plaintext histogram of Clock; (**e**) Ciphertext histogram of Lena; (**f**) Ciphertext histogram of Cameraman; (**g**) Ciphertext histogram of Peppers; (**h**) Ciphertext histogram of Clock.

**Figure 8 entropy-23-00291-f008:**
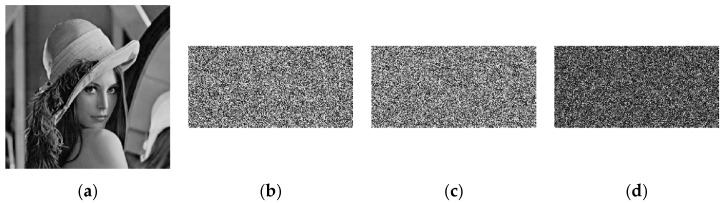
Key sensitivity analysis of encryption process. (**a**) Plaintext image; (**b**) Ciphertext image encrypted using the original key; (**c**) Ciphertext image encrypted using the modified key; (**d**) Difference of the images in (**b**,**c**).

**Figure 9 entropy-23-00291-f009:**
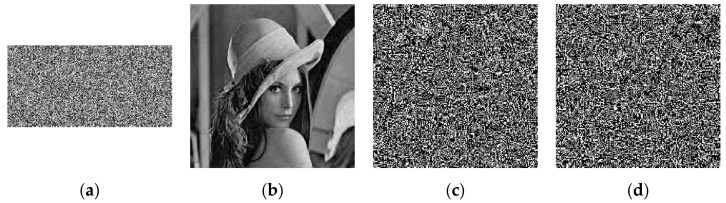
Key sensitivity analysis of decryption process. (**a**) Ciphertext image; (**b**) Image decrypted using the correct key; (**c**) Image decrypted using modified key $x_{1}{}^{\prime }(0)$; (**d**) Image decrypted using modified key $x_{2}{}^{\prime }(0)$.

**Table 1 entropy-23-00291-t001:** *ApEn* results of different chaotic sequences.

Hyperchaotic System	Input Parameters	ApEn
Lorenz	N = 2000, m = 2, r = 0.2 SD	0.6608
Chen	N = 2000, m = 2, r = 0.2 SD	0.8171
Ref. [43]	N = 2000, m = 2, r = 0.2 SD	1.3698
Ours	N = 2000, m = 2, r = 0.2 SD	1.9814

**Table 2 entropy-23-00291-t002:** NIST test results.

Randomness Test		*P*-Value						Passing Rate
		$x_{1}$	$x_{2}$	$x_{3}$	$x_{4}$	$x_{5}$	$x_{6}$	
Frequency		0.6171	0.7856	0.5864	0.8540	0.2428	0.6833	992/1000
Block Frequency (m = 128)		0.8351	0.6974	0.6662	0.4928	0.7466	0.9059	996/1000
Runs		0.2119	0.4752	0.0252	0.3281	0.1740	0.4390	998/1000
Longest Run of One’s		0.1186	0.7177	0.4391	0.2148	0.4522	0.7123	991/1000
Matrix Rank		0.5887	0.0755	0.0323	0.3321	0.2269	0.0520	992/1000
FFT		0.3098	0.6424	0.3675	0.2701	0.5232	0.7768	996/1000
Non-Overlapping Template (m = 10)		0.5837	0.5488	0.4926	0.6498	0.4611	0.1333	997/1000
Overlapping Template (m = 10)		0.2357	0.7061	0.6839	0.2893	0.7191	0.9533	992/1000
Universal		0.6438	0.1190	0.3546	0.5336	0.0976	0.1152	996/1000
Linear Complexity (m = 1000)		0.4632	0.2193	0.3235	0.3964	0.1824	0.6857	994/1000
Serial test	*P*_value1	0.2547	0.7748	0.0215	0.7236	0.0951	0.1991	997/1000
	*P*_value2	0.2369	0.7315	0.6471	0.2807	0.4727	0.1073	992/1000
Approximate Entropy (m = 3)		0.1768	0.4481	0.5260	0.1330	0.5561	0.7177	996/1000
Cumulative Sums test	Forward	0.5267	0.1637	0.3939	0.2647	0.5815	0.6057	995/1000
	Reverse	0.6054	0.0977	0.2861	0.1585	0.2784	0.4893	993/1000
Random Excursions		0.3556	0.5075	0.5326	0.2780	0.7651	0.6296	997/1000
Random Excursions Variant		0.5673	0.5386	0.4412	0.4818	0.3967	0.7104	993/1000

**Table 3 entropy-23-00291-t003:** PSNRs (dB) of different compression/encryption schemes.

Image	CR	Ours	Ref. [52]	Ref. [16]	Ref. [36]
Lena	0.75	36.21	29.56	29.22	33.25

**Table 4 entropy-23-00291-t004:** Correlation coefficients of adjacent pixels in different images.

Direction	Lena		Cameraman		Peppers		Clock	
	Plaintext	Ciphertext	Plaintext	Ciphertext	Plaintext	Ciphertext	Plaintext	Ciphertext
Horizontal	0.9701	−0.0023	0.9545	−0.0064	0.9625	0.0044	0.9741	0.0017
Vertical	0.9432	0.0051	0.9352	0.0141	0.9533	−0.0056	0.9579	−0.0149
Diagonal	0.9169	0.0124	0.9068	0.0016	0.9258	0.0098	0.9424	−0.0068

**Table 5 entropy-23-00291-t005:** Correlation coefficients of the encrypted Lena image compressed using different algorithms.

	Horizontal	Vertical	Diagonal
Lena	0.9701	0.9432	0.9169
Encrypted Lena	−0.0023	0.0051	0.0124
Ref. [29]	−0.0065	0.0073	0.0042
Ref. [55]	0.0597	−0.0766	0.0083

**Table 6 entropy-23-00291-t006:** Information entropy of plaintext and ciphertext images.

	Compression Ratio	Image			
		Lena	Cameraman	Peppers	Clock
Information Entropy	0.25	7.9889	7.9892	7.9886	7.9878
	0.5	7.9951	7.9953	7.9946	7.9943
	0.75	7.9968	7.9966	7.9963	7.9965

**Table 7 entropy-23-00291-t007:** UACI and NPCR values obtained using the proposed algorithm.

Image	NPCR (100%)	UACI (100%)
Lena	99.6080	33.4561
Cameraman	99.6143	33.4589
Peppers	99.6106	33.4697
Clock	99.6071	33.4721

**Table 8 entropy-23-00291-t008:** Encryption and decryption time of the proposed scheme (in s).

	CR	Lena	Cameraman	Peppers	Clock
Encryption time	0.25	0.4115	0.4125	0.3958	0.4064
	0.5	0.5028	0.4959	0.4761	0.5030
	0.75	0.6127	0.5933	0.6067	0.6169
Decryption time	0.25	3.5099	3.0761	3.1410	3.1358
	0.5	6.4591	6.2996	6.3754	6.3499
	0.75	10.5292	10.6391	10.4168	10.5386

## Data Availability

Data sharing not applicable.

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
