# Peer review of "Image Encryption Scheme with Compressed Sensing Based on a New Six-Dimensional Non-Degenerate Discrete Hyperchaotic System and Plaintext-Related Scrambling"

_entropy, 2021, doi:10.3390/e23030291_

Round 1
Reviewer 1 Report
Dear Authors,
It was really interesting to me to review your manuscript containing such a thorough technical and mathematical study. The presented research can make a significant contribution to the multimedia data processing and storage. However, I could not recommend this article for publication in its current form due to the following reasons:
1. Please clarify what is meant by "more complex" in the sentence: In addition, the chaotic dynamics behavior is more complex. How one can calculate this property?
2. By what principle were the values of the coefficients of system (1) chosen? Please describe the rule used. Can you replace these ratios with custom parameters? By varying these values, it is possible to increase the calculated values of the Lyapunov exponents. And it will also allow us to consider the rich behavior of this system on bifurcation diagrams.
3. The article is framed carelessly. For example, the notation ApEn is not introduced.
4. For what parameters of the Lorenz and other chaotic systems were the values in Table 1 calculated? This is important because the resulting ApEn value can be highly dependent on them.
5. How many sequences were considered for NIST testing? What was them length? What are the proportions of the past sequences?
6. Why was SHA-256 chosen to generate hashes, if other algorithms are known, including those based on chaotic systems, which may be faster and more secure? For example, adaptive chaotic maps have recently been proposed. Please review these techniques and make at least a theoretical comparison.
Reviewer 2 Report
It was my pleasure to read and review this comprehensive manuscript. Please, find my comments below.
1. "However, these encryption algorithms focus on information security and do not take into account the high image data redundancy, strong correlation between pixels, and other characteristics of such images." (p.1)
It is very strong statement. Indeed, the considered algorithms possess some known shortcomings. However, there are many other recently proposed chaos-based image encryption methods, e.g. using minimal digital chaotic systems, maps with adaptive symmetry and other approaches that are less vulnerable to chaos degradation and pixel correlation effect. Please, revise the introduction accordingly to improve the state-of-the-art section.
2. The proposed chaotic system seems to be really complex computationally. Image processing performance highly depends on image resolution. Moreover, the comparison with existing non-chaos-based image encryption techniques should be discussed. I believe many of existing generators can also successfully pass NIST being much simpler algebraically.
3. Image encryption is a known basis for video stream encryption. This raises again the question of computational efficiency. The section "Simulation and Performance Analysis" provides only qualitative performance analysis, considering compression quality, losses etc. However, the compression time and its dependence on image size are of great interest here. Please, perform some efficiency tests of your approach in comparison with existing encryption techniques
4. No experimental check for chaos degradation is performed to verify the robustness of the proposed system, especially for long-term simulation case.
5. The proposed system relies on a precisely set matrix of coefficients. How the short machine word length will affect your algorithm? Is it possible to implement this encryption in embedded hardware, e.g. using a 32-bit platform? Will its superiority preserve?
Nevertheless, my overall impression is high and I can recommend the paper for publication after moderate revisions.
Reviewer 3 Report
Authors designed a combining approach for image encryption. Paper has practical value.
It has a logical structure, all necessary sections. Paper is technically sound. Experimental section is very good!
The proposed approach are logical, results are clear.
Authors should take into account the following issues:
- Introduction section should be extended using existing encryption techniques.
- It would be good to add clear point-by-point the main contributions in the end of the Introduction section
- It would be good to add the reminder of this paper in the end of the Introduction section
- I can't find time performance of the proposed technique
- A lot of references are outdated. Please change it using 3-5 years old papers in high-impact journals.
Reviewer 4 Report
Paperd deals with an image encryption task. Authors have developed a new approach for solving the stated task
Paper has a scientific novelty and practical value.
The title of the article is clear and logical, the annotation fully reflects the essence of the research.
The work is well structured, contains a graphic illustrations for better understanding the proposed approach.
Suggestions:
1. Introduction section should extend this section using overview of the different cryptografic techniques. In particular authors can analyze the disadvantages of the methods based on the combining use of the elements of RSA algorithm and different mathematical aparatus (10.1016/j.procs.2019.11.057, 10.1016/j.procs.2019.11.059, 10.1016/j.procs.2019.11.043, 10.1016/j.procs.2019.08.089)
2. The advantage of this article is the large number of experimental studies performed by the authors. All of them are aimed at determining the effectiveness of the proposed encryption scheme from different angles. However, it would be good to see the time performance of the proposed scheme, or to indicate the need for such research in the future (perspective of the future research).
3. The conclusions should be extended by 1) obtained in the paper results 2) limitations of the proposed approach 3) prospects of the future works.
Round 2
Reviewer 1 Report
Dear Authors,
Thank you for taking my recommendations into account. I still have a couple of comments about the manuscript, after correcting which I consider it possible to publish this article.
1. For reliable results with NIST tests, more than 1000 sequences need to be considered, while the used sample is much smaller. Please, See 4.2.1 Section in NIST recommendations
Thus, it is necessary to test NIST for a larger sample.
2. My previous comment about theoretical comparison was not a suggestion to complicate the described scheme. In the current version of the article there is a comparison with continuous systems inclding Lorenz system et al., The simulation of such systems involves the use of integration methods, which is slower than discrete chaotic maps in ecryption tasks. I propose to make a comparison with faster and more efficient algorithms, for example, adaptive chaotic maps, which were used both to generate hashes and in stream ciphers to generate pseudo-random sequences in chaos-based cryptography. Thus, please add such comparison and the relevant references as the discussion. This will help to better understand the advantage of the proposed approach in comparison with the fastest known solutions.
Reviewer 2 Report
The choice of the encryption techniques for comparison in the Introduction is still pretty weird.
The provided complexity analysis revealed that the significant time is required for decompression \ decoding of the image with the proposed method. This is a major drawback, especially in comparison with existing techniques based on simpler (e.g. adaptive symmetric or minimal chaotic) maps which are not discussed in the paper.
I believe for larger pictures the delay will be more significant. This prevents the proposed technique for broad use in real problems, especially in video encryption. I recommend the authors pay more attention to this aspect.
Nevertheless, I am generally satisfied with the answers and therefore can recommend the paper for publication after minor revisions.
Reviewer 3 Report
Paper can be accepted
Author Response
We thank you for your thoughtful suggestions and insights. The manuscript has benefited from these insightful suggestions.
Reviewer 4 Report
Paper can be accepted in the current form
Author Response

(The authors gave the same response as above.)
